# Micronuclei in Circulating Tumor Associated Macrophages Predicts Progression in Advanced Colorectal Cancer

**DOI:** 10.3390/biomedicines10112898

**Published:** 2022-11-11

**Authors:** Dimpal M. Kasabwala, Raymond C. Bergan, Kirby P. Gardner, Rena Lapidus, Susan Tsai, Mohammed Aldakkak, Daniel L. Adams

**Affiliations:** 1School of Graduate Studies, Rutgers University, New Brunswick, NJ 08901, USA; 2Creatv MicroTech Inc., Monmouth Junction, NJ 08852, USA; 3Fred & Pamela Buffet Cancer Center, University of Nebraska Medical Center, Omaha, NE 68198, USA; 4Greenebaum Cancer Center, School of Medicine, University of Maryland, Baltimore, MD 21201, USA; 5Department of Surgery, Medical College of Wisconsin, Milwaukee, WI 53226, USA

**Keywords:** micronuclei (MN), advanced colorectal cancer, cancer associated macrophage like cells (CAMLs), circulating stromal cells (CStCs), liquid biopsy, genotoxins, molecular stress, DNA damage

## Abstract

Micronuclei (MN) are fragments of damaged nucleic acids which budded from a cell’s nuclei as a repair mechanism for chromosomal instabilities, which within circulating white blood cells (cWBCs) signifies increased cancer risk, and in tumor cells indicates aggressive subtypes. MN form overtime and with therapy induction, which requires sequential monitoring of rarer cell subpopulations. We evaluated the peripheral blood (7.5 mL) for MN in Circulating Stromal Cells (CStCs) in a prospective pilot study of advanced colorectal cancer patients (n = 25), identifying MN by DAPI+ structures (<3 µm) within the cellular cytoplasm. MN+ was compared to genotoxic induction, progression free survival (PFS) or overall survival (OS) hazard ratios (HR) over three years. MN were identified in 44% (n = 11/25) of CStCs, but were not associated with genotoxic therapies (*p* = 0.110) nor stage (*p* = 0.137). However, presence of MN in CStCs was independently prognostic for PFS (HR = 17.2, 95% CI 3.6–80.9, *p* = 0.001) and OS (HR = 70.3, 95% CI 6.6–752.8, *p* = 0.002), indicating a non-interventional mechanism in their formation. Additionally, MN formation did not appear associated with chemotherapy induction, but was correlated with tumor response. MN formation in colorectal cancer is an underlying biological mechanism that appears independent of chemotherapeutic genotoxins, changes during treatment, and predicts for poor clinical outcomes.

## 1. Introduction

Colorectal cancer (CRC) is the third most common cancer malignancy and second most leading causes of cancer-related deaths in the US, with a reported estimated 151,030 new cases of CRC, and an estimated 52,580 deaths predicted in 2022 [1]. While there are a number of non-invasive screening techniques developed to aid in diagnosing the disease earlier, i.e., stool based Cologuard^®^ test, CEA in plasma, and CT colonography [2], the only method for identifying CRC is through a colonoscopy, or polypectomy, when possible [3]. Roughly ~20% of CRC patients are diagnosed with metastatic CRC (mCRC), with another 25% of patients diagnosed with local disease that will later develop metastases [4]. Patients diagnosed with mCRC have extremely low survival rates, with fewer than 20% of patients surviving 5 years [4]. Currently, treatment for advanced CRC is relatively limited, but often includes surgery or radiation therapy, after initial treatment with first-line systemic neoadjuvant chemotherapy consisting of intravenous 5-flourouracil/leucovorin oxaliplatin (FOLFOX), or Irinotecan (FOLFIRI) if response with oxaliplatin fails [4,5]. Overall, the development of new systemic therapies for CRC and the poor response rates to FOLFOX/FOLFIRI has been obstructed by the CRC’s ability to become treatment resistant, proliferate and spread rapidly [4,5,6]. Therefore, a better understanding of the mechanisms involved in developing this sub-clonal resistance is crucial to distinguishing patients with more aggressive CRCs.

Small portions of distinct DNA separated from a cell’s primary nucleus, but within the cytoplasmic area, were discovered over 100 years ago and defined as Micronuclei (MN). These MN have long been thought of as a sign of genomic instability and a key biomarker of cancer [7,8], originating from a cellular repair mechanism in response to acentric chromosomal and chromatid fragments or whole chromosomes which failed to segregate properly during mitotic division [7]. MN can form naturally, due to the chromosomal separation errors during cellular division, particularly during anaphase of mitosis, though MN may form other biological phenomena, i.e., double-stranded DNA breaks, impaired DNA repair [9,10]. Damaged DNA lesions when unable to repair can lead to the formation of acentric chromosomes and consequently, MN [11]. However, environmental genotoxic stressors, such as environmental carcinogens, UV radiation, or chemotherapy can also result in similar chromosomal errors and subsequent MN formation by aiding in the rupturing of the nuclear envelope allowing for MN escape from the primary nucleus [8,11,12,13,14]. In both instances, natural or environmental, MN are the terminal result of the cellular repair of genomic errors which allow for cell survivability by the excision of fatal genomic errors that otherwise would be lethal to cells. DNA damage that occurs in MN induces chromosomal rearrangements, a process called chromothripsis which then allows these MN formations to serve a purpose in the pathogenesis of cancer [10]. These chromosomal rearrangements occur simultaneously, potentially resulting initiation of uncontrolled cellular growth and cancer in an “all-at-once” fashion [10]. In addition to MN within cancer cells, the appearance of MN and genome instability has also been described in circulating WBCs (cWBCs) which has been shown to be an indication of cancer evolution and survivability [9]. Likely, MN in cWBCs results in the chronic activation of the innate immunity, via presence of MN in the cytosol of immune cells, resulting in a pro-tumorigenic proinflammatory response [15,16]. In cancer, it has been hypothesized that MN formation resulting from therapeutic genotoxins, such as chemotherapies and radiation, represent the survival mechanism that tumor cells undergo to sustain therapeutic interventions and may represent cancer cell sub-populations are of somatic evolutionary significance and inheriting resistant to these therapies.

The presence of MN in the cytoplasm of certain populations of cWBCs has been shown to indicate both cancer presence and cancers with higher rates of progression and with poorer outcomes [11,17]. This has largely been hypothesized resulting from a person’s overall cellular response to an exposure to genotoxins, not specific to tumorigenesis itself [18]. However, the origin, specific typing of, and the underlying biology of MN positive cWBC subpopulations remains unknown. Recently, MN were superficially observed in a small cWBC population of cancer emanating circulating stromal cells (CStCs), identified as a cancer associated macrophage-like cells (CAMLs), which appeared only in the peripheral whole blood of patients with solid tumor malignancies, including colon and other gastrointestinal cancers [19,20,21]. CAMLs were described as giant polyploid myeloid cells with a great diversity of nuclei structures [19,22,23,24,25]. Interestingly, MN formation were superficially observed in CAMLs, though the biological and clinical meaning of MN bodies within CAMLs was not evaluated. Using a population of patients who had undergone standard chemotherapeutic treatments, we sought to determine the relationship of MN formation in circulating cells in patients as it relates to genotoxic chemotherapy, and their relationship to tumor resistance by tracking and measuring patient progression and survival.

## 2. Material and Methods

### 2.1. Peripheral Blood Collection

A prospective single-blind study was initiated to quantify the presence of MN within CAMLs in advanced CRC patients. Twenty-five CRC patients with advanced disease, stage III (n = 8) or metastatic stage IV (n = 17), had 7.5 mL whole peripheral blood collected at either time of diagnosis and before the induction of any therapy (n = 13), or after the failure of a prior treatment regimen and before the start of a new systemic treatment (n = 12). Patients were all recruited after pathological confirmation of advanced colorectal adenocarcinoma from University of Maryland Baltimore (UMB), Oregon Health & Science University (OSHU), Medical College of Wisconsin or Quest Diagnostics. Inclusion criteria required patients with advanced colorectal adenocarcinoma above the age of 18. Minors were not permitted in this study. All blood samples and patient information were collected with written informed consent and with Institutional Review Board (IRB) approval, or through clinical trial NCT04504942 also with written informed consent and IRB approval. Blood samples were anonymized and shipped within 96 h of blood draw for sample processing. Patient clinical information was not shared or unblinded until the completion of the study.

### 2.2. Liquid Biopsy

Blood samples (7.5 mL) were collected into CellSieve™ vacutainer tubes and filtered using CellSieve™ Microfilters to isolate cancer-associated macrophage-like cells (CAMLs), circulating tumor cells (CTCs), and epithelial to mesenchymal transition cells (EMTs) [19,23]. Specifically, blood was filtered to isolate CAMLs, CTCs and EMTs through 7 µm pores to separate from white blood cells (WBCs) and red blood cells (RBCs). Microfilters were washed with phosphate buffering saline solution (PBS) and placed onto the center of a filtration device. 7.5 mL of whole blood was prefixed with equal amounts of Prefixation Buffer for 15 min and was then filtered through the microfilter to collect circulating cells. After filtration, the microfilter was treated with postfixiation buffer, permeabilization buffer, and stained with antibodies mixture including FITC (Cytokeratin) and Cy5 (CD45), all as previously described [19]. Filters were washed with 10 mL of PBS = 0.1% Tween-20 (PBST), 2 mL of PBS, and finally mounted with 4′,6-diamidino-2-phenylindole (DAPI).

### 2.3. Cell Imaging and MN Analysis

Analysis of cells was completed using an Olympus BX54WI Fluorescent microscope with a Carl Zeiss AxioCam (Zen 2.3 Blue edition, Carl Zeiss Microscopy GmbH, White Plains, NY, USA) and was used to image all cell populations by a trained cytologist. A Zen2011 Blue 3.0 (Carl Zeiss) was used to process the images and further analyze the cells. CAMLs and CTCs are detected in this process by their phenotypic expression of CD45, Cytokeratins 8, 18, 19, and DAPI, as previously described [19,26] (Figure 1). CAMLs were identified by their enlarged polynucleated nucleus (ranging from 14 to 64 microns) and by their giant cellular bodies (roughly 30 to 300 microns in length.) Other tumor-associated cells such as circulating tumor cells (CTCs) and epithelial-to-mesenchymal transition cells (EMTs) can be identified through these filtration methods, however, they were not included in the analysis of MN, as MN were not found in these cell populations. After identification of the cell populations, DAPI positive events were identified by the Zen2011 Blue (Carl Zeiss) and presented to a cytotechnician. MN were defined as small circular DAPI positive structures found within the cytoplasmic region of each cell, but distinct and separate from the primary nucleus.

### 2.4. Statistical Analysis

Analysis was conducted using MATLAB R2021b from all patients. Cox proportional hazard regression plots and hazard ratios (HRs) were determined by log-rank analysis with significance defined as a *p*-value < 0.05. Cox proportional hazard regression was also used to determine the univariate and multivariate hazard ration with aa statistical analysis’s threshold of p-value < 0.05, using MATLAB R2021b. Univariate analysis was individually run using all available clinical parameters from all patients (Appendix A). Student *t*-tests were performed in determining *p*-values in comparing MN presence in various subsets of this cohort. Kaplan–Meier compared the presence and absence of MN in CAMLs for progressive-free survival (PFS) and overall survival (OS). Progression was defined as the time of baseline blood draw, prior to new therapy induction, until the growth of the tumor by CT/PET scan as defined by RECIST 1.1, or until new disease.

## 3. Results

Blood samples were collected from 25 colorectal patients. The mean age of the cohort was 51 years with a range between 27 and 92 years and interquartile range of 45 and 61 years (Table 1 and Appendix A). Patients were diagnosed with either locally advanced (stage III, n = 8) or advanced metastatic colorectal cancer (stage IV, n = 17). Over half the patient population (n = 13) received prior systemic treatment, while the other half of the population was newly diagnosed treatment naïve (n = 12). After the baseline blood draw, the treatment cohort received either FOLFOX (n = 17), FOLFIRI (n = 4), or single agent inhibitors (i.e., leronlimab or cetuximab) (n = 4). All patients were diagnosed with adenocarcinoma of the colon, with one patient later subtyped after start of study as appendix cancer.

Within the patient population, there was a total presence of 288 CAMLs detected at baseline, i.e., first blood draw, with a positivity rate of 84% in all patients (n = 21/25). CAMLs, which are giant polyploid immune cells, were identified by their large size, morphological characteristics, expression of Cytokeratin or CD45 positivity, and enlarged multinucleated structure (Figure 1). CAMLs were observed with distinct MN within the cytoplasm, and with budding-like structures of possible micronuclei from the primary nucleus, possibly indicating the process of MN formation (Figure 1e), though MN were not observed in other tumor associated circulating cells such as CTCs or EMTs. These CAML MN were observed as small fragments, average size of ~2 µm (Figure 1e), and were found in 44% (n = 11) of the CAMLs but did not appear to correlate with the overall baseline CAML number, *p* = 0.715 (Figure 2a). This was despite the currently MN hypothesis which suggests that innate immune activation is related with these specific genomic events (Figure 2a). We then compared MN presence in patients who had been previously treated with chemotherapy (i.e., FOLFOX) (n = 13) and chemotherapy naïve patients (Figure 2b). Looking at patients who had MN presence in CAMLs, 52% of these patients have received a previous systemic therapy. Conversely, patients who did not have MN formation in CAMLs, 36% of these patients have received previous systemic therapies, and there appeared to be no association between MN presence and chemotherapy induction (*p* = 0.110). Analysis of CRC staging found MN formation in 25% (n = 2/8) of stage III patients and 52% (n = 9/17) in patients diagnosed with stage IV, indicating that MN frequency has no significant relationship to metastatic spread, *p* = 0.137 (Figure 2c). Overall, analysis found that MN formation is a somewhat rare occurrence, ranging from 0–33 individual MN in all CAMLs, and occurring only in ~8% (Figure 2d). Additionally, the majority of CAMLs (n = 708) in this study did not present MN positivity (Figure 2d), with many MN positive cells (~9%) found with ≥1 MN (n = 68 CAMLs), and ~5% of all analyzed CAMLs having 2 or more MN events (n = 35 CAMLs).

Numerous studies have shown that MN presence in certain populations of cWBCs are indicative of cancer presence and patients with worse clinical outcomes including in CRC, though the origin and specific subtype of these cells is unknown [11,17]. As CAMLs are a specific circulating stromal cell population that has been shown to be highly prognostic for patient outcomes [19,20,23,24,25,27,28,29], we evaluated the CAML population based on MN presence. After primary analysis of MN presence based on numerous clinical variables, progression free survival (PFS) and overall survival (OS) outcomes were monitored for 36 months after the baseline blood draw. It was found that patients with MN in CAMLs had faster rates of progression, with MN presence associated with a median PFS of 5 months vs. a median PFS of over 36 months in patients without MN (Figure 3a). Upon further analysis over 36 months, MN presence in CAMLs was found to be a significant predictor of PFS (HR = 17.2, 95% CI 3.6–80.9, *p* = 0.0014) (Figure 3a). In addition, it was observed that patients with MN CAML positivity had faster rates of mortality, with MN presence associated with a median OS of 7 months vs. a median OS of over 36 months in patients without MN (Figure 3b). Similarly, MN presence in CAMLs was determined to be a significant predictor of OS over 36 months (HR= 70.3 95% CI 6.6–752.8, *p* = 0.0027) (Figure 3b).

To understand the temporal change in MN presence as it relates to treatment induction, 8 patients volunteered for multiple follow up blood draws after baseline sampling, of which 2 patients gave multiple blood samples at every treatment cycle for ≥6 months, or until diagnosis of progressive disease. The two patients further studied are not representative of the entire population, however, serve as examples in understanding the impact MN presence has on patient outcomes throughout time. One patient (Patient A) had been previously treated for metastatic disease with recent progression to the lungs while on Xeloda maintenance after completion of previous treatment regimens (Figure 4a). At start of therapy with FOLFOX and after 1 full cycle, there were no MN identified. After 3 cycles of FOLFOX, the average MN increased to 0.5 MN/CAML, and the patient then temporarily stopped therapy due to an infection. At cycle 4, a PET scan found an increase of +17% which was defined as stable disease, that appeared to parallel with a 60% increase in MN count. Three additional follow up blood draws saw consistent increases of MNs/CAML to a total of 9 MN at cycle 5, 8 MN at cycle 6, and 45 MN at cycle 7. At cycle 6, new progressive disease was identified by PET/CT scan and FOLFOX was re-started. This progression appeared to coincide with the increasing MN formations within CAMLs, though further clinical information was not available. In a second case study, Patient B had an initial blood draw with high numbers of MN (3.1 MN/CAML) and had progressive disease while on FOLFOX. A new single agent CCR5 inhibitor, leronlimab [29], was started which coincided with a dramatic drop in MNs after 1 cycle of therapy (Figure 4b). After 4 cycles of therapy, a scan confirmed a partial response of −39% in all target lesions. Additional blood draws found an increase in MNs at cycle 6 and 7, which was then associated with an 11% increase from the prior scan. Based on the lesion increase, the patient was started on FOLRIRI along with leronlimab, which saw a MN drop to 0 MN/CAML and the patient was found to have stable disease, with no changes in lesions observed. However, at cycle 10, the MN then increased to an average of 0.195 MN/CAML and a subsequent PET/CT was found to have a +58% increase in target lesions and new pulmonary nodules. Overall, the changes in MN presence within blood CAMLs appeared to track in real time the changes in the tumor response to new therapies, which coincided with similar findings to standard of care PET/CT.

## 4. Discussion and Conclusions

In this pilot study, we evaluated the frequency of MN formation in giant circulating polyploid immune cells which originate from primary tumor sites, i.e., circulating stromal cells, in 25 advanced stage CRC patients. We evaluated the associations between CAML presence, genotoxic exposure, and metastatic spread to the formation of MN in CAMLs by comparing the frequency of MN formation in CAMLs from the entirety of CRC population. Overall, the frequency of MN in CAMLs was rare, ~8% of all CAMLs examined (Figure 2). This is much lower than the rate of MN formation listed in prior publications, which averaged 61 MN per 186 baseline peripheral blood lymphocytes in CRC, and other solid tumor cancers [30]. However, this rarity is likely a result of the cancer specificity of CAMLs, which are an established cancer specific cWBC subtype, versus other groups studying MN formation with general blood lymphocytes which are more ubiquitous [11,17,19,27,30,31]. As MN presence has been previously associated with activation of the innate immune system, we examined the potential effects of MN formation on the number of CAMLs in circulation [16,20,21,29], finding no significant correlation between the presence of MN and the number of CAMLs found in circulation (Figure 2). This may suggest that although MN formation may be involved in the recruitment of immune components, the MN pathway observed here did not appear to affect the recruitment/dissemination of CAMLs from the primary tumor. Though as the patient cohort size was limited in this study, the immune activation effects of MN formation on CAMLs may not have been pronounced and future more in-depth studies involving a more comprehensive look at immune recruitment, MN, and CAMLs would be required. Further, as the GAS-STING pathway has been implicated in these activating effects of MN, looking at the expression of various downstream markers of this pathway in conjunction with MN formation in CAMLs may be of interest [16,32].

Next, we examined the effects of chemotherapy and stage on the formation of MN in CAMLs, finding that there was no significant difference in MN formation in CAMLs between treatment naïve and patients previously treated with systemic chemotherapy (*p* = 0.110) (Figure 2b). This points to an alternative origin of the MN formation in CAMLs, other than the introduction of genotoxins, which was the original hypothesis of this study. This finding suggests, as CAMLs are aberrant giant tumor associated cells, their MN formation might not be related to an external environmental process, but internal genomic instability related to pathogenesis itself [19,25,33]. This might be a suggestive of a potential genomic mechanism, shedding damaged DNA from the primary nucleus in response to DNA damage from uncontrolled/incomplete replication cycles, though this hypothesis was not evaluated this study design and must be further evaluated [7]. Further, as hypothesized by Adams et al. 2014, an alternate hypothesis to the origin of these cytoplasmic DNA MN could be the phagocytosis of cancer cells and neoplastic materials by CAMLs at the primary tumor site [19,34]. Lastly, we looked at the prevalence of MN formation in CAMLs between stage III and IV disease. Once again, this was a null result as there was no significant difference the two stages of disease (Figure 2c). While it appears that MN formation could not be used to distinguish between late stage and metastatic CRC, MN were found in both stages of CRC, suggesting the potential prevalence in earlier stage disease. Interestingly, as CAMLs are a biomarker in early stages of many types of cancer, the formation of MN may be an added parameter in cancer identification and increased risk of more aggressive disease subtypes [20,23,24,26,27,28].

Further investigation MN formation must be conducted to gain better understanding on MN presence in larger cohorts consisting of various cancer malignancies. MN form as a resistance mechanism, as a result of genotoxic or mutagenic events, such as the induction of chemotherapy and is indicative of cancer evolution characterized by genomic rearrangements. Interestingly, these rearrangements include aberrations such as translocations, mutations, and fusion events, depending on the quantity and type of gene damage. Translocations are commonly found in cancer cells and are possible precursors to oncogenes responsible for malignancies [10]. Because of MN formations, change in chromosomal compositions or aneuploidy can trigger the development of translocations. Additionally fusion events, fusions between cell types, may also cause denaturation of DNA, thus expressing the need for repair and MN formation [9,32]. Translocations and fusion events that take place are likely to damage DNA which is likely to undergo DNA repair mechanisms during these unstable genetic events resulting in the formation of MN. Genome chaos will likely arise from various molecular mechanisms which producing chromosomal instability thus inducing evolutionary survival mechanism to repair DNA [9]. Since not all cancer cells are likely to undergo cellular death during chemo-radiation therapy, surviving cancer cells are likely to proliferate and grow thus possibly aiding in tumorigenesis and cancer recurrence [7,11,31]. In vivo experiments have shown that cells with MN are often chemo-resistant, indicating a biological pathway for cellular survival and cancer evolution [9]. It is hypothesized that these genotoxic therapies used on that cancer patients receive (i.e., chemotherapy) may are likely to result in DNA repair mechanisms that leads to formation and tumor resistance [7,11,31]. Overall, MN within the cytoplasm of cells is indicative of a chromothripsis event, possibly associated with chemoresistance and may indicate a quantifiable tumor cell survival mechanism during pathogenesis [7,10,11,31].

In this initial pilot study, it appears the MN formation in CAMLs may predict for progression and mortality (Figure 3), and further, tracking the formation of MN in CAMLs during the course of treatment appears to correlate to responsiveness of the patient to the various treatments (Figure 4). While there is a clear need to expand this study and validate these preliminary findings, it would be of interest to take on a more thorough study examining MN in CRC. MN have been hypothesized to be potential cause of chromothripsis, which can lead to ‘all-in-one’ mutations via chromosomal rearrangements leading to the formation of cancer [10,11]. CAMLs have been shown to be prevalent in early stage gastrointestinal cancers so examining these cells for MN presence may be beneficial in early stage cancer detection and cancer subtyping [19,20,21,23,26]. The examination of MN formation in CAMLs in early stage CRC might allow for discriminating more aggressive disease and target these disease subtypes with more aggressive treatment interventions. Further of interest, is identifying which chromosomes are more involved in the formation of MN, and if MN formation is associated with other oncogenic events such as exosome signaling and mitochondrial apoptosis [35,36]. Lastly, the cytosolic DNA has been shown to upregulate certain immune markers such as PD-L1 [37]. This could suggest that an examination in the expression of certain immune blockade pathways on CAMLs with MN (i.e., PD-L1, STING, etc.) could provide an interesting potential immunotarget to study. Irregardless, the role of MN in CAMLs appears to have a clear relationship to the aggressiveness of CRC and clinical outcomes. Thus, it is logical to develop a better understanding of the biological development of MNs on tumor associated CAMLs and further evaluated both the underlying biology of CAML MN and if it is useful interventionally for patients with solid malignancies.

## Figures and Tables

**Figure 1 biomedicines-10-02898-f001:**
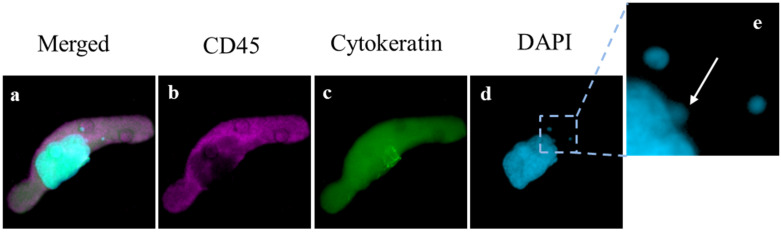
Image of CAML with Micronuclei. (**a**–**d**) single CAML with enlarged nucleus (blue) diffuse Cytokeratin (green) and CD45 (violet) indicating cell structure (Box = 85 μm). (**e**) Enlarged portion of cell with one micronuclei budding from the primary nucleic mass (white arrow) and two separate distinct micronuclei also within the cell cytoplasm (Box = 20 μm).

**Figure 2 biomedicines-10-02898-f002:**
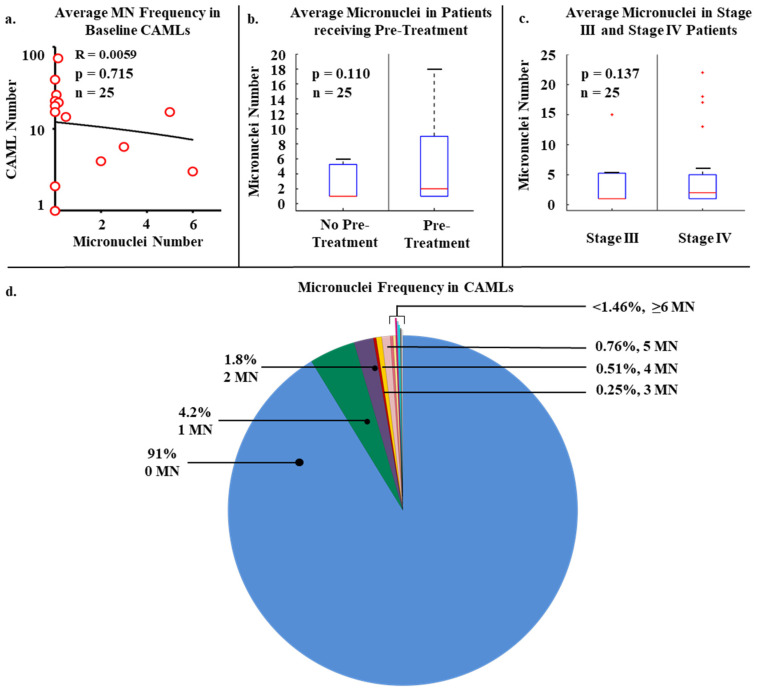
Micronuclei presence in the various clinical parameters. (**a**) Linear regression plot of the relationship of CAML number versus micronuclei number in each patient at baseline sampling. (**b**) Micronuclei presence based on patients currently on chemotherapy (i.e., genotoxin) or patients that were newly diagnosed and treatment naive. (**c**) MN presence based on non-metastatic or metastatic spread. (**d**) Micronuclei frequency in all CAMLs (n = 776). 91.0% of CAMLs (blue) were negative for micronuclei, while only 9% were positive for at least 1 micronuclei.

**Figure 3 biomedicines-10-02898-f003:**
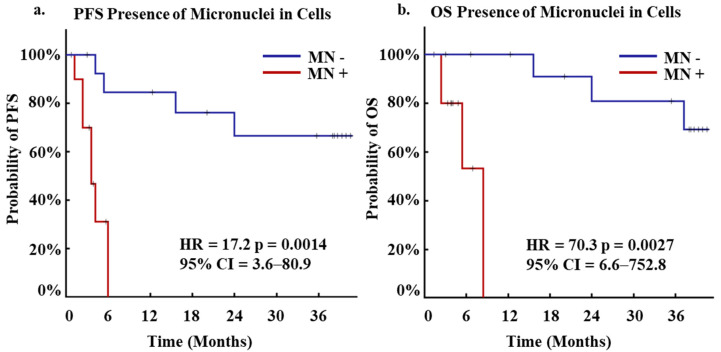
Kaplan-Meier of PFS and OS for Micronuclei positivity in patients at first blood draw. (**a**) Kaplan-Meier of PFS for patients who presented positivity for micronuclei (red, N = 11) or negative (blue, N = 14). (**b**) Kaplan-Meier of OS for patients who presented positivity for micronuclei (red, N = 11) or negative (blue, N = 14).

**Figure 4 biomedicines-10-02898-f004:**
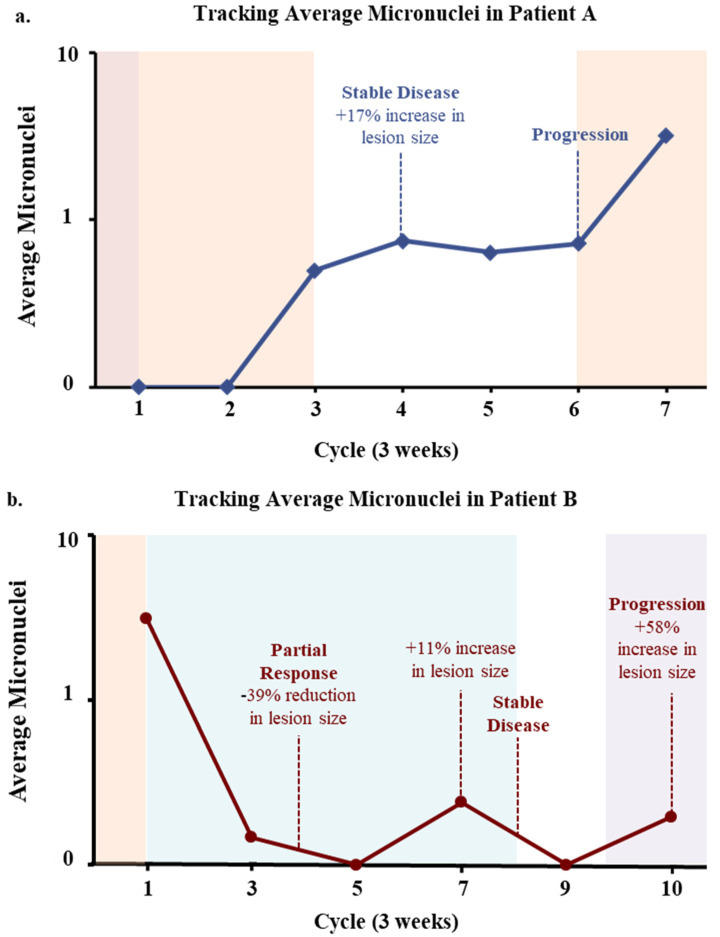
Case Studies Tracking Average Micronuclei in Subsequent Blood Draws. (**a**) Patient A. After progression on maintenance therapy (Xeloda, red), FOLFOX (orange) was started. MN were found to increase after 4 therapy cycles which correlated to an increase in tumor (+17%). Therapy was halted due to an infection. MN then increased further correlating with finding of new lesions at cycle 6. FOLFOX was then restarted, and MN continued to increase, the patient then dropped from study. (**b**) Patient B. A patient with progressive disease on FOLFOX was started on a single agent CCR5 inhibitor (Leronlimab, blue), which was followed by a decrease in MN correlating with a reduction (−39%) in tumor size. MNs then increased slightly, which correlated with a slight increase in tumor size (+11% and a possible new lung lesion) at which point FOLFIRI (purple) was started. After FOLFIRI induction, a drop in MN was seen which correlated with stable disease.

**Table 1 biomedicines-10-02898-t001:** Clinical demographic for patients.

Patient Demographics	n = 25
**Age (median)**	51
Age IQR, Range	45–61, 27–92
**Gender**	
Male	17 (68%)
Female	8 (32%)
**Race**	
Caucasian	13 (52%)
African American	1 (4%)
Hispanic	1 (4%)
Asian	1 (4%)
Unknown	9 (36%)
**Stage**	
Non–metastatic (II–III)	8 (32%)
Metastatic (IV)	17 (68%)
**Prior Treatment**	
Pre–Treatment	13 (52%)
No Pre–Treatment	12 (48%)
**Current Treatment**	
FOLFOX	17 (68%)
FOLFIRI	4 (16%)
Other **	4 (16%)
**Histology**	
Adenocarcinoma *	25 (100%)

* 1 patient: identified as adenocarcinoma but diagnosed with appendix cancer, ** Single agent inhibitor (i.e., cetuximab, Leronlimab, etc.).

## Data Availability

The data presented in this study are available within the manuscript.

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
