# Peer review of "Micronuclei in Circulating Tumor Associated Macrophages Predicts Progression in Advanced Colorectal Cancer"

_biomedicines, 2022, doi:10.3390/biomedicines10112898_

Round 1

Reviewer 1 Report (Previous Reviewer 2)

No further comments to the authors.

Author Response

Thank you for your review. Quirina Xue, the editor for my submitted manuscript wrote back to me and said that I should "resubmit the manuscript to the system and we will send it to academic editor to make the decision" since there was a lack of major revisions to reply to. Thank you so much for your help during the review process of this manuscript. 

Reviewer 2 Report (Previous Reviewer 1)

 OK

Author Response

Thank you for your review. Quirina Xue, the editor for my submitted manuscript wrote back to me and said that I should "resubmit the manuscript to the system and we will send it to academic editor to make the decision" since there was a lack of major revisions to reply to. Thank you so much for your help during the review process of this manuscript. 

Reviewer 3 Report (New Reviewer)

The manuscript by Dimpal et al., demonstrates the micronuclei in circulating tumor associated macrophage (CAMLs) predicts increased cancer risk and progression in advanced colorectal cancer. In this article, authors observed that micronuclei, fragments of damaged nucleic acids from the cell nuclei in the circulating stromal cells of colorectal cancer patients are co-related with the survival (PFS and OS) and hazard ratios. However, these were not corelated with either genotoxic therapies or metastasis. The article is well written and organized.

Specific comments:

·         In fig 2C, stage III micronuclei number does not have an error bar. Please correct it.

Author Response

Reviewer 3: “In fig 2C, stage III micronuclei number does not have an error bar.”

  1. Thank you so much for the comments and suggestions. We have mended our error and have added an error bar in Figure 2c. Stage III graph. We have also included the mended Figure 2 in the final manuscript.

This manuscript is a resubmission of an earlier submission. The following is a list of the peer review reports and author responses from that submission.

Round 1

Reviewer 1 Report

The manuscript by Kasabwala and colleagues is extremely interesting. The use of specific cancer associates cells, rather that circulating lymphocytes seem to be promising for the screening of early stages of cancer, colon cancer in this case. The authors have correctly reported their findings under the perspective of a pilot study and as such several hypotheses are presented and discussed.

I have only minor comments. The first is dealing with the large heterogeneity of patient’s age. Age has been reported as the main determinant in MN frequency in peripheral blood lymphocytes and at a lesser extent in exfoliated buccal cells (see many papers by Fenech and coll), and therefore the presence of MN in different age-classes should be evaluated, as well as its potential effect on OS and PFS.

As regards methods, the authors should say something more about Cox models used to estimate HRs. Were they univariate or some covariates were added ? In this last case what was the goodness of fit ?

Finally, the part I did not like of the paper was the discussion of the two cases with repeated MN measurement. The approach is interesting, under the perspective of monitoring cancer progression, However, the results do not seem very convincing and the interpretation of figures is forced in some instances. I suggest to substantially reduce the emphasis on these results.

Author Response

Reviewer 1: “Age has been reported as the main determinant in MN frequency in peripheral blood lymphocytes and at a lesser extent in exfoliated buccal cells (see many papers by Fenech and coll), and therefore the presence of MN in different age-classes should be evaluated, as well as its potential effect on OS and PFS.”

  1. Although age has been seen as a noticeable parameter in other works, our analysis of different age-classes was not found to be significantly correlated to MN, and thus we had not included it in the previous data sets. However, we have now included these results comparing the MN presence against age of patients as a Supplementary Figure 1.
  2. Additionally, the univariate/multivariate analysis of age for PFS and OS has also now been included in a Supplementary Table 1.

Reviewer 1: “As regards methods, the authors should say something more about Cox models used to estimate HRs. Were they univariate or some covariates were added ? In this last case what was the goodness of fit ?”

  1. Cox proportional univariate regression analysis was conducted against all known clinical variables individually, but most were not found to be significant. However, a multivariate analysis was run on all significant variables. We have now included the results of the individual univariate analysis as well as the results of the multivariate data analysis in a Supplementary Table 1.
  2. Additionally, we have expanded the materials and methods section as requested to include the following verbiage, “Cox proportional hazard regression were used to determine the univariate and multivariate hazard ration with a statistical analysis’s threshold of p-value <0.05, using MATLAB R2021b. Univariate analysis was individually run using all available clinical parameters from all patients. Multivariate analysis was run using all available significant univariate variables.”

Reviewer 2 Report

The present manuscript by Kasabwala et al evaluates if any co relation exists between Micronuclei (MN) formation in circulating cancer associated macrophage-like cells (CAML) and genotoxic effect of chemotherapy, tumor resistance, metastatic progression and survival in Colorectal cancer patients. Though the study certainly is of interest the authors did not find any association between the above-mentioned phenomenon and the study has been done in a small cohort of patients. Also, the current manuscript does not represent a significant enough conceptual advance beyond previously published literature about MN or CAMLs formation. Overall, it's unable to show any new direction. Thus, in my opinion the study does not qualify to be published in the stature of journal like IJMS in its current form.

Author Response

Reviewer 2: “…the authors did not find any association between the above-mentioned phenomenon and the study has been done in a small cohort of patients.”

  1. We agree with reviewer 2 that as a proof of principle pilot study, the cohort of patients is smaller than a full training/validation set study. However, as reviewer 1 mentioned, we report these findings as a “perspective of a pilot study”. Hopefully after publishing these findings, we can expand and validate our results in a larger cohort of patients.
  2. We also agree with the reviewer that there was no significant correlation between MN and other clinical parameters of these patients. This was a surprise to us as well, and was discussed in detail in our discussion. However, the MN in CAMLs was correlated with worse clinical outcomes suggesting that CAML MN is a previously unstudied phenomenon worthy of further evaluation.

Reviewer 2: “Also, the current manuscript does not represent a significant enough conceptual advance beyond previously published literature about MN or CAMLs formation.”

  1. We find this critique somewhat puzzling. We are not aware of any biological nor clinical research on MN in CAMLs, much less of any clinical study evaluating their clinical relationship. For example, the most recent review on CAMLs published in March of 2022 by Dr. Tang & Adams (citation 24) made no mention and cited no works on MN in CAMLs.